# Brief Communication: An Autonomous UAV for Catchment-Wide Monitoring of a Debris Flow Torrent

Fabian Walter[1,2], Elias Hodel[2,1], Erik S. Mannerfelt[2,1], Nicolas Ackermann[3], Kristen Cook[4], Michael Dietze[5,4], Livia Estermann[6], Michaela Wenner[2,1], Daniel Farinotti[2,1], Martin Fengler[7], Lukas Hammerschmidt[7], Flavia Hänsli[6], Jacob Hirschberg[1,6], Brian McArdell[1], and Peter Molnar[6]

[1]Swiss Federal Institute for Forest, Snow and Landscape Research WSL, 8903 Birmensdorf, Switzerland
[2]Laboratory of Hydraulics, Hydrology and Glaciology, ETH Zurich, 8093 Zurich, Switzerland
[3]Swiss Federal Railways SBB, 3000 Bern, Switzerland
[4]GFZ German Research Centre for Geosciences, 14473 Potsdam, Germany
[5]Faculty of Geosciences and Geography, Georg-August-University Göttingen, 37077 Göttingen, Germany
[6]Institute of Environmental Engineering, ETH Zurich, 8093 Zurich, Switzerland
[7]Meteomatics AG, 9014 St. Gallen, Switzerland

**Correspondence:** Fabian Walter (fabian.walter@wsl.ch)

**Abstract.**

Debris flows threaten communities in mountain regions worldwide. Combining modern photogrammetric processing with autonomous unoccupied aerial vehicle (UAV) flights at sub-weekly intervals allows mapping of sediment dynamics in a debris-flow catchment. This provides important information for sediment disposition that pre-conditions the catchment for debris flow occurrence. At the Illgraben debris-flow catchment in Switzerland, our autonomous UAV launched nearly 50 times in the snow-free periods in 2019-2021 with typical flight intervals of 2-4 days, producing 350-400 images every flight. The observed terrain changes resulting from debris flows exhibit preferred locations of erosion and deposition, including memory effects as previously deposited material is preferentially removed during subsequent debris flows. Such data are critical for the validation of geomorphological process models. Given the remote terrain, the mapped short-term erosion and deposition structures are difficult to obtain with conventional measurements. The proposed method thus fills an observational gap, which ground-based monitoring and satellite-based remote sensing cannot fill as a result of limited access, reaction time, spatial resolution, or involved costs.

## 1 Introduction

Water discharge peaks can mobilize sediments in steep torrents, which subsequently move in the form of debris flows toward channel outlets. Warning systems often rely on rapid detection of debris flows via monitoring precipitation, ground unrest or flow depth (e.g. Badoux et al., 2009). However, repeated catchment surveillance is also important for assessing debris flow hazards. In this way, water-damming deposits from previous debris flows or landslides can be identified, whose breaching may be particularly difficult to predict since it is not related to meteorological parameters (Godt and Coe, 2007). Similarly, repeated digital elevation models (DEMs) can reveal temporal exhaustion of sediments available in debris flow source areas.

This "supply limitation" temporarily lowers the debris flow hazard, in contrast to sudden slope failures whose deposits in torrent beds suddenly increase the hazard (Bovis and Jakob, 1999). Such variations in sediment availability may explain why rainfall thresholds tend to perform poorly in terms of warning (Cannon et al., 2008; Kean et al., 2012; Gregoretti et al., 2016; Rengers et al., 2016). However, the spatial coverage and the temporal resolution (typically on the order of tens of km$^2$ and days to weeks, respectively) needed to reliably monitor an entire catchment requires costly surveillance flights or time demanding site visits.

Here we introduce a new approach for monitoring sediment changes in catchments that are prone to debris flows. Using an autonomous unmanned aerial vehicle (UAV) performing flights at intervals as low as a few days, we generate time series of DEM differences for a Swiss torrent. We employ a recently developed photogrammetric processing scheme to identify terrain changes in the hillslope-channel area with decimeter precision, showing erosion and deposition patterns related to debris flows and to lateral slope failures. We propose to integrate the system into multisensing monitoring approaches to optimize the assessment of debris flow hazards in otherwise difficult-to-access mountainous regions.

## 2 Study Site: Illgraben, Switzerland

This study focuses on the Illgraben torrent in Switzerland's Canton Valais (VS), which drains a 9 km$^2$ catchment (Figure 1) and produces around 5 debris flows per year on average reaching the channel outlet at the Rhône River (Badoux et al., 2009). With little sediment discharge outside debris flow episodes, Illgraben delivers annually some $10^5 \, m^3$ of material to the Rhône (Hirschberg et al., 2021a). Mobilized into debris flows during intense summer precipitation, sediment deposits within the upper channel sections are supplied from frost-weathering slope failures (Bennett et al., 2013; Hirschberg et al., 2021b, a). Probabilistic modelling indicates that sediment supply limitations affect the formation of large debris flows, although there are no direct observations to confirm this (Bennett et al., 2014; Hirschberg et al., 2021a).

Entrainment and positive feedback between sediment motion in the channel and on lateral slopes result in debris flow volumes that may exceed $10^5 \, m^3$ and thus the volumes of individual rockfalls and landslides feeding the channel (Schlunegger et al., 2009; Berger et al., 2011; Burtin et al., 2014). As an exception, in 1961 a rock avalanche filled the upper channel section with $3.5 \times 10^6 \, m^3$ of sediments (Gabus et al., 2008). A series of 30 check dams were constructed in response to this event downstream of the deposits to hold back debris flow discharge into the Rhône River and to stabilize the channel (Lichtenhahn, 1971). Since the check dam construction, debris flows have rarely overtopped the channel banks and have caused little damage. However, a risk to tourists traveling on popular hiking trails crossing the channel remains. Additionally, many parts of the Susten village, which lies on Illgraben's debris fan, are threatened by events with long return periods (Badoux et al., 2009). As a result, an alarm system signals in-torrent detection of mass movements at the Illgraben mouth (Figure 1; Badoux et al., 2009). Additional instrumentation near the channel outlet includes flow depth gauges, a force plate for instantaneous flow weight measurements, automatic cameras, infrasound microphones, and seismometers (e.g., McArdell et al., 2007; Marchetti et al., 2019; Schimmel et al., 2018; Chmiel et al., 2021).

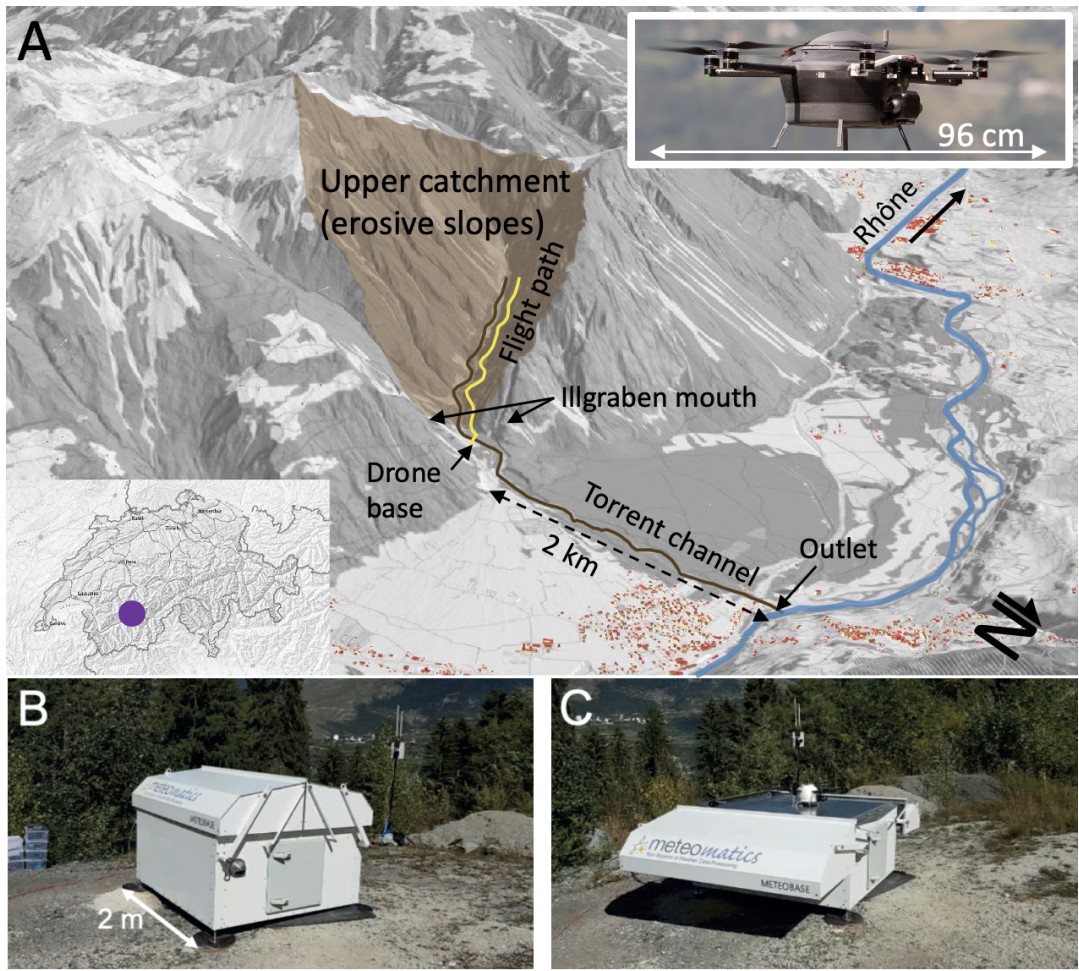

**Figure 1.** Overeiw of Illgraben's upper catchment area, where high erosion on lateral slopes deposits material in the torrent channel (brown line), which mobilizes to debris flows. Yellow line denotes the UAV flight path. Blue line represents the Rhône river and map inset indicates Illgraben's location within Switzerland (purple dot). Photograph inset shows the hexacopter Meteodrone MM-670 with a size of 96 cm measured between two most distant propeller tips. Red symbols depict buildings. (B-C) UAV base with open (C) and closed (B) lid. The hexacopter used for acquiring the airborne imagery is visible in panel C, at the center of the take-off and landing area. Hillshade in Panel A from Swiss Federal Office of Topography swisstopo.

Repeated topographic surveys generating DEMs have been used to study the controls on erosion and deposition by debris flows at Illgraben: a terrestrial laser scanner has been used over a $300\,m$ long reach of the main Illgraben channel for 14 events occuring between 2007 and 2009 (Schürch et al., 2011) and a UAV over a $3\,km$ long channel section mainly on the fan, which was flown before and after six debris flows in 2018 and 2019 (de Haas et al., 2020). These studies provided highly resolved topographic data between individual debris flows and provided insights into the roles of channel geometry, check dams and debris-flow characteristics in erosion and deposition processes. For example, debris flows tend to erode at locations where the previous event was depositional, and to deposit where previous events were erosional (de Haas et al., 2020). To study variability in sediment production, four aerial images recorded over 2008 and 2009 were sufficient to identify a downslope-directed sediment cascade at the seasonal scale (Berger et al., 2011). Sediment dynamics were also studied over decadal time scales (42 years in total) but at a coarser temporal resolution (6-20 years). Aerial images showed, for example, an increase in the Illgraben erosion rate from the 1980s, likely due to decreased snow cover and enhanced weathering (Bennett et al., 2013). While these studies were helpful in describing patterns of sediment supply from hillslopes and its relation to sediment yield, the mass movement initiation mechanism remains difficult to identify. Similarly difficult is the assessment of sediment budgets at the event scale, since some eroded areas may be masked in the aerial images and since the reconstructions from older images are affected by uncertainties of up to 5 m (Bennett et al., 2013).

## 3   Autonomous UAV

To monitor sediment dynamics in the upper catchment, where access on foot and in-torrent instrumentation is limited, we deployed an autonomous UAV near the Illgraben mouth for several months during summers 2019-2021 (Figure 1, Table 1). The system was developed by the company "Meteomatics" (www.meteomatics.com) and consists of (i) an autonomous UAV home-base called the Meteobase and (ii) a hexcopter called the Meteodrone (Figure 1). The Meteobase (type MM-B4) has a lid that opens automatically, and is of size $1.6 \times 1.8 \times 1.3$ m (length, width, height) when closed and of $3.3 \times 1.8 \times 1.3$ m in its open state. This creates an effective landing area of $1.5 \times 1.5$ m. The Meteobase was deployed in combination with Meteodrones of type MM-670B or MM-670C. The Meteodrone MM-670B weighs about $4.8$ kg and has a diameter of less than 1 m. The UAV is equipped with a safety parachute system and redundant Inertial Measurement Unit, Global Navigation Satellite System (GNSS) and compass instruments. The GNSS receiver accepts real-time kinematic (RTK) corrections via the Global System for Mobile Communications (GMS) network, and serves as the main navigation instrumentation. For landing, the UAV relies on RTK GNSS in combination with an infrared and vision-based tracking system. The Meteobase was connected to a 220 V power supply and recharged the Meteodrone automatically through a specific charging port. The latter automatically connects to the UAV from a hole in the landing platform, which is sealed prior to and after charging. One fully charged battery yields about 30 min of flight time. Batteries were charged with a current of 20 A at 24 V whose power was provided to the Meteobase via a continuous external line power supply. A power generator has been employed in previous installations of the Meteobase. A combination of solar-panels and batteries would also be feasible but has yet to be implemented. The external line power supply also feeds the other electric consumers, such as the air conditioning unit used for climatizing the Meteobase's

interior during very hot or cold days. The Meteobase also acts as an operational relay between Meteodrone and operator, which remotely supervises the flights as demanded by regulations. The Meteobase also ensures that procedures such as charging, data upload and download, UAV positioning on the landing platform, or climatizing, are performed automatically. Meteodrone and Meteobase communicate through radio connection, whereas the base and the remote operator communicate via a combination of 4G GSM and Local Area Network internet connections.

As payload, the UAV carried a Yuneec E90, 20 megapixel, nadir-oriented photocamera with an electronic shutter and a 1-inch Complementary Metal Oxide Semiconductor sensor. The camera view included mainly the torrent channel and therefore lateral slopes were only surveyed near the channel. Pictures were taken every 2 seconds leading to an overlap of 70 to 80 %. Synchronization of RTK geolocation and camera shutter was not implemented for technical reasons. This synchronization is planned in future deployments. Other technical challenges included limited GNSS and GSM reception, as well as limited durability of components which required replacement. These were parts that were susceptible to environmental conditions and which needed adaptions or a replacement for more robust versions. One example is the compass, which failed at the beginning of the campaign due to direct sunlight radiation and developing heat and was replaced with a more reliable version. Fixing such simple yet critical technical glitches was the reason why more maintenance was necessary in the beginning of the project, compared to the end of the project when inspections were carried out about once every two months. In general, the Meteodrone requires maintenance on the motors after 150 flight hours, exchange of the battery after 150 recharging cycles and parachute replacement after 12 months. In total, the setup allowed for 41 autonomous flights during July-October 2020 and July-August 2021 after a test period in 2019 (Table 1).

In Switzerland, remote UAV operation beyond visual line of sight (BVLOS) falls into a specific flight regulation category and requires a Specific Operation Risk Assessment (SORA)[1]. The SORA was requested from the Federal Office of Aviation (FOCA) and contained both the remote operation surveyed from a control center in St. Gallen (Switzerland) and autonomous BVLOS flights into the Illgraben. At present, FOCA requires redundancy for flight-critical instruments and sensors, as well as strict geofencing. In addition, a surveillance webcam was installed next to the Meteobase, enabling the operator to check for a cleared landing area.

Interventions of the operator were limited to Abort Missions, e.g. due to strong wind, or changing weather conditions. The operator has different options to intervene, depending on the situation. In case the UAV is still capable of flight, the operator can abort the mission or return to launch. The operator was also allowed to hold the flight or descent to a safe altitude. The parachute rescue system is either launched automatically upon recognition of a problem by the onboard systems or at any time by the operator. This means that also during the starting and landing phase the operator can "kill" the system should it be necessary to prevent harm. The operator can observe the landing and starting site through the surveillance camera, ensuring nobody is in the immediate vicinity. We decided against a fenced base station for this project, but this may be appropriate in the future to protect the Meteobase and Meteodrone from vandalism.

Catchment-wide flights refer to the along-channel section between the Meteobase and the head of the Illgraben channel. This extent was covered by 6 km-long round-trip flights taking approximately 20 minutes each. The UAV flew at 100 m altitude

---

[1]https://www.bazl.admin.ch/bazl/en/home/good-to-know/drohnen/wichtigsten-regeln/bewilligungen/sora.html, last accessed 05.04.2022

above the torrent channel with a speed of 5-7 m/s, taking between 350 and 400 photographs along its way. The average ground sampling distance amounted to ca. 10 cm per pixel.

| | 2019 | 2020 | 2021 |
|---|---|---|---|
| Dates (dd.mm) | 01.10.-17.10. | 03.06.-21.08. | 03.05.-02.08. |
| Number of catchment-wide flights | 9 | 10 | 17 |
| Number of sub-catchment flights | 4 | 8 | 1 |
| Typical flight interval (days) | $\sim$2 | $\sim$3 | $\sim$4 |
| Average number of images per flight | 407 | 313 | 308 |

**Table 1.** Autonomous UAV operation at Illgraben for the years 2019, 2020, and 2021.

## 4 Photogrammetric Processing & Results

Only 2 usable Ground Control Points were collected in the accessible section of the channel, 500-1000 m upstream of the UAV base. This means that the GNSS coordinates acquired by the UAV were the only reliable georeferencing information. Initially, the autonomous imagery was processed by using the built-in functionalities of the software Agisoft Metashape version 1.7.0. Images were aligned for each survey individually, using their full resolution, rolling shutter compensation, an image location accuracy of 20 cm, and otherwise default parameters. However, the accuracy of the RTK GNSS positions proved to be insufficient: tie-point residual errors never converged to sub-pixel levels, which indicates a faulty camera model, most likely resulting from poor georeferencing (James et al., 2017). We attribute this inaccuracy to the insufficient synchronization between the internal clock of the camera and the clock of the GNSS receiver, leading to incorrect matches with the GNSS track (c.f. Girod et al., 2017). To lower the georeferencing and tie point residual errors to reasonable levels, a more advanced alignment technique was therefore needed.

We opted for the "co-alignment" approach, proposed by Cook and Dietze (2019) for processing surveys for change detection. In this workflow, the images from two or more surveys are pooled during Agisoft Metashape's image alignment and model optimization processing steps. After the model geometry is set, the dense clouds are then calculated separately for each survey. The identification of common tie points in stable areas visible in photographs from different surveys results in a model that fixes the different surveys with a common geometry. While this approach does not improve the accuracy of global georeferencing, the high comparative precision between co-aligned surveys makes the approach effective for constraining geomorphic change (Hendrickx et al., 2020; Watson et al., 2020). Autonomous surveys are particularly suited for this type of co-alignment approach, as the photographs show a high consistency in orientation, altitude, and location from survey to survey. For our imagery, calculated elevation differences of stable terrain indicated a height error of 0.2 m for the co-alignment processing. The processing of one flight took about four hours on a 12-core 3 GHz Intel Zeon central processing unit without graphics processing unit acceleration.

Figure 2 shows an example of a selection of DEM differences from flight pairs that include one or more debris flows, which reached the channel outlet. At the foot of lateral tributary gullies, upstream erosion during winter months left deposits within the channel, with thicknesses of more than 2 m (Section A in Figure 2). Subsequent debris flows preferentially eroded these deposits, which is a manifestation of a "memory effect" that had previously been observed behind Illgraben's check dams on the debris fan (de Haas et al., 2020). In the upper catchment part surveyed in the present study, check dams are also subject to this memory effect (Section B in Figure 2). Also apparent are typical lateral levee deposits (e.g. Johnson et al., 2012), and erosion within the channel flow center (Section C in Figure 2).

## 5  Discussion and Conclusion

The UAV-based monitoring of a debris flow catchment proposed here allows detection of changes in sediment deposits on meter scales or less. Larger slope failures, significantly changing sediment supply and debris flow hazards can thus be accurately detected. We argue that this approach can also be applied to map terrain changes in other contexts, like fluvial erosion during flood events, snow avalanches, or shallow and deep-seated landslides. Indeed, all of these mass movements may demand timely reaction with post-event intervention.

This investigation has shown that quantitative information on catchment-wide sediment dynamics can be obtained on timescales of hours, i.e. on timescales that are only constrained by UAV battery charge and flight times. This fills a critical gap left by costly airborne sensing and satellite-based methods, which have multi-day return periods.

Although the main channel was surveyed far upstream, debris-flow initiation areas and the triggering mechanism could not be identified unambiguously. It is clear, however, that lateral tributaries deliver significant amounts of sediments to the main channel, where they are temporarily stored. The re-mobilization of these deposits may contribute to debris-flow formation and subsequent supply-limitations (Berger et al., 2011). Currently, our flight planning and the nadir-looking camera view do not allow quantification of the individual tributary contributions to debris flow volumes and dynamics. However, it is possible to approximately date relevant tributary activity and study its controls, which will be subject of future systematic research using the entire present data set. We also envision the application of our method for constraining geomorphological models that describe sediment movement in response to short-term meteorological forcing.

Our application of an autonomous UAV leverages recent success of photogrammetry for monitoring geomorphological processes. UAV-based photogrammetry combined with the structure-from-motion algorithm (Smith et al., 2016), which was also used in our co-alignment approach (Cook and Dietze, 2019), offers the unique advantage of mapping extended portions of changing terrain. This is a valuable alternative to DEM differences derived from terrestrial laser scanning (TLS), which has also been used in debris flow catchments to study erosion patterns in relation to debris flow dynamics (Dietrich and Krautblatter, 2019), process domains of debris flow initiation (Staley et al., 2014) and channel response to climate signals (Bonneau et al., 2022). Whereas topography derived from UAV-based photogrammetry and TLS agrees within a few decimeters (Cook, 2017), the former method has the advantage of mapping poorly accessible terrain where TLS equipment cannot be installed.

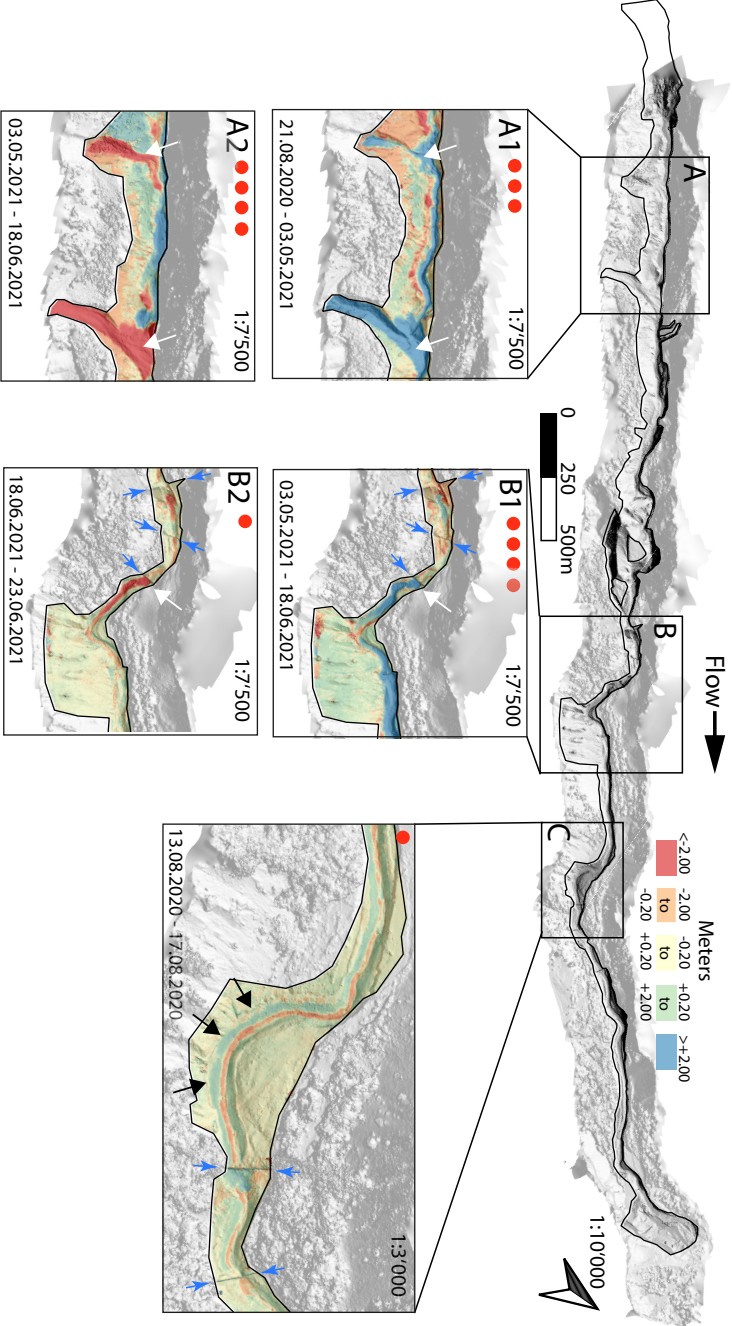

**Figure 2.** Hillshade of Illgraben section covered by autonomous UAV flights. Black solid outlines enclose areas mapped by DEM differencing. Zoom boxes focus on erosion and deposition features, as shown by DEM differences. Times of flight pairs are indicated at the bottom of each zoom box. Red dots represent the number of debris flows occurring between flights, as registered by the debris flow observatory of the Swiss Federal Institute for Forest, Snow and Landscape Research WSL (Badoux et al., 2009). Boxes A and B illustrate memory effects, with erosion concentrated in areas of previous deposition (white arrows). A and B show the cases for tributary deposition within the channel and debris flow deposition behind a check dam, respectively. Box C highlights lateral levee formation and in-channel erosion (black arrows). Color map gives erosion (-) and deposition (+) in meters. Note that scaling factors are relative among zoom boxes. In Boxes B1, B2 and C, thin blue arrows indicate the location of check dams.

Despite its advantages, UAV-based photogrammetry has difficulty in resolving terrain changes for certain types of surfaces, in particular vegetation cover (Cook, 2017). This disadvantage, however, may be negligible for active debris flow catchments, whose sediment dynamics allow for little or no sustained vegetation growth. Moreover, our application of co-alignment reduces the dependence on ground control points which are also difficult to obtain in poorly accessible terrain. Consequently, the main constraints of our proposed sediment dynamics monitoring with an autonomous UAV are the dependence on a reliable power and internet connection for the UAV base as well as GNSS signal and radio connection between the base and the UAV.

In the future, autonomous UAV operation can be linked to other sensor systems: The UAV could be sent to map runout and damage immediately after an event, which could be detected by seismic or infrasound sensors, for example. These latter methods have the advantage of a large radius of sensitivity. However, accurate event location and volume estimates like UAV-derived DEM differences provide, are often unavailable for seismic and infrasound monitoring. In the spirit of an "Internet of Things (IoT)" approach, the UAV system could be integrated within autonomous multi-sensing platforms that leverage the strengths of individual sensor components. The aftermath of the 2017 rock-avalanche at Piz Cengalo, Switzerland, underlined the urgent need for such post-event monitoring: within 1-2 weeks, unstable rock avalanche deposits subject to high pore pressures produced 15 debris flows destroying parts of the village of Bondo (Walter et al., 2020). Such rapid secondary effects of the rock avalanche were not expected but in the future could be monitored and warned against with a quickly deployable autonomous UAV.

Rapid technological developments and increasing sensor coverage targeting rapid mass movements are currently preparing the ground for autonomous monitoring and warning systems for Alpine hazards. For our specific case, BVLOS flight permissions still required a human to follow the UAV operation from a remote location. Apart from legal constraints, this type of surveillance was not necessary from a technical and operational point of view. We thus anticipate that it is only a question of time until the presented technology will find its way into IoT monitoring solutions for natural hazards in Alpine terrain.

*Data availability.* UAV images are archived a WSL and access can be granted by the authors.

*Author contributions.* FW, NA, MW and DF planned the autonomous UAV operation at Illgraben. EH, EM, KC and MD tested conventional and co-alignment processing of the photogrammetric data. LE, FH and EH applied the co-alignment processing to all available images and together with FW, BM, JH and PM interpreted the results in terms of torrential activity. MF and LH lead the UAV deployment and operation.

*Competing interests.* The authors declare no competing interests.

*Acknowledgements.* This project was financed by WSL, SBB, Meteomatics and Amberg Loglay. We thank the Burgerschaft Leuk for permission to install and operate the UAV base at Illgraben and the FOCA for granting BVLOS permission. Pierre Huguenin and Mauro Marty helped with ground control points.

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
