# Peer review of "Brief Communication: An Autonomous UAV for Catchment-Wide Monitoring of a Debris Flow Torrent"

_EGUsphere, 2022_

## Author Comment (AC1)

Referee's comments are in dark blue.

**General comments**

This "Brief communication" presents a novel technique that enables the monitoring of sediment dynamics in remote terrains. It combines photogrammetric processing with UAV and was tested in the Illgraben debris-flow torrent, located in Switzerland. The topic is perfectly fitting with the ones proposed by NHESS and the contents are relevant for researchers and practitioners. I recommend the publication of the ms after some minor revisions.

- I suggest adding some information on recent studies applying UAV (or TLS) in torrential or fluvial areas (not related to Illgraben).

  Agreed. We will provide more information about recent techniques to map and monitor sediment dynamics in torrential/fluvial areas other than Illgraben. Depending on available space, we may have to focus on UAV's and keep the discussion on TLS very short.

- The (preliminary) results, described between L125 and 131, should be extended and placed in a separate section. The results are brilliant and deserve a longer description. Not only related to the sediment dynamics, but also on basic (more technical) information like the pixel size of the DEM, which is missing.

  Agreed. We will discuss the results in the following contexts:

  - potential for identifying debris-flow initiation areas
  - role of lateral sediment inputs for sediment dynamics in main channel
  - sediment supply limitations

  We will furthermore provide DEM pixel size.

If the above two points are not possible due to space problems, try to reduce other parts of introduction or discussion-conclusions.

- The description of locations like Illgraben mouth, channel outlet, upper catchment, head of the Illgraben channel, catchment outlet are not always clear. The authors may simplify them and add the most important ones in Figure 1A.

  Agreed. We will annotate Figure 1 and reduce closed terminology (e.g., as appropriate, only use "head of the Illgraben channel" OR "upper catchment").

**Specific comments:**

L30: "DURING debris flows" is not clear. The surveys were before and after debris-flow events, weren't they?

A subtle difference but certainly correct! We will change the wording accordingly.

L73: width not with

OK.

L84-85: not totally clear, which was finally used in Illgraben (LAN or/and GSM).

We will clarify.

L89: "1-inch Complementary Metal Oxide Semiconductor sensor" only expert may understand it. Please clarify.

This is a technical specification, which we deemed of interest to photogrammetry experts. The acronym "CMOS" is more widely used, but since we do not use this term again, we assume that we have to spell it out.

L125: Figure 2 (not 1).

OK.

Figure 2: Improve the design or layout (e.g. rotate the zoom boxes and make them larger). Clearly indicate the locations of the check dams.

OK. Zoom boxes can be enlarged, but we feel that they should be oriented the same way as the overview. Check dams will be indicated in the zoom boxes.

L140: You only mention flight time, but it would also be interesting having some information on the time consumption of the photogrammetric processing.

Agreed. We will provide processing time estimates.

---

## Author Comment (AC2)

Referee's comments are in dark blue.

In this 'brief communication', Walter et al. introduce a new technique for (semi-) autonomous high-resolution monitoring of sediment dynamics in remote alpine terrain based on UAV remote sensing and photogrammetry. The presented system consists of a hexacopter and a base station that facilitates automatic recharging and acts as an operational relay between the UAV and a remote operator, who is in charge of the flight monitoring. From a technical and operational point of view, a remote operator would not be necessary within this framework, but supervision by a human is still mandatory for (autonomous) UAV missions beyond visual line of sight. The feasibility of the approach has been successfully demonstrated at the Illgraben debris-flow catchment in Switzerland and paves the way for (semi-)autonomous UAV-based monitoring in other contexts and terrains.

In addition to the comments of Reviewer 1, I have some (rather technical) remarks and questions that might be of interest for some readers:

1) A limited durability and replacement of some system components is mentioned in the text (L93-94). Which components were less reliable or failing and how often had the base station to be serviced during the summer months? The maintenance aspect would be especially important for autonomous operations in even more remote locations where the frequent replacement of components is difficult.

We plan to provide the following explanations (or an abbreviated version thereof): The mentioning of limited durability mostly refers to components which were susceptible to environmental conditions and which needed adaptions or a replacement for more robust versions. One example is the compass, which failed at the beginning of the campaign due to direct sunlight radiation and developing heat and was replaced with a more reliable version. This example shows that more maintenance was necessary in the beginning of the project, compared to the end of the project. Towards the project end, inspections were done about once every two months. In general, the Meteodrone requires maintenance on the motors after 150 flight hours, exchange of the battery after 150 recharging cycles and the parachute after 12 months.

2) Had the remote operator/observer in the control centre in St. Gallen ever to intervene during the three-year period? Has the remote operator/observer full control over the UAV and what would happen in the case of an (unlikely) emergency (e.g. bird attack or curious people/animals approaching the base station during the survey or landing process)? I assume the base station is fenced. Is this correct?

We plan to provide the following explanations (or an abbreviated version thereof): Interventions of the operator were limited to *Abort Missions*, e.g. due to strong wind, or changing weather conditions. The operator has different options to intervene, depending on the situation. In case the UAV is still capable of flight, the operator can abort the mission or return to launch. The operator was also allowed to hold the flight or descent to a safe altitude. In case the UAV is not capable of flight, it is equipped with a parachute rescue system. The rescue system is either launched automatically upon recognition of a problem by the onboard systems or at any time by the operator. This

means also during the starting and landing phase the operator can "kill" the system should it be necessary to prevent harm. The operator can observe the landing and starting site through a surveillance camera, ensuring nobody is in the direct area. We decided against a fenced base station for this project, but this may be appropriate in the future.

3) Can you say anything about the precise and autonomous landing of the copter: is the RTK GNSS in combination with a vision-based tracking system used for this purpose?

We will add the following explanations: For landing, the copter relies on RTK GNSS in combination with an IR and vision-based tracking system.

4) Does the base station rely on an external power supply or is it connected to solar panels and a power station (not shown in Fig. 1)?

We plan to provide the following explanations (or an abbreviated version thereof): In the Illgraben setup, the system was relying on line-current and continuous external power. However, the power could be supplied by other means. An external power generator has been employed before, but so far, no solar panels. A solar-panel battery combination would be feasible.

5) I'm aware that the selected manuscript type has a strict page limit, but I am missing a brief discussion on similar approaches/applications (i.e. semi-autonomous UAV monitoring) in the geosciences.

We will add some text on this question in the introduction (see also comment by Referee 1).

Wording: Unoccupied Aerial Vehicle is increasingly used in the (geo)scientific community as a more neutral term for UAV (Joyce et al., 2021: https://doi.org/10.3390/drones5010021) and I therefore suggest to adopt it.

Thank you for this remark, we will discuss it with the photogrammetry experts of the study and may change the terminology accordingly.

---

## Author Comment (AC3)

Referee's comments are in dark blue.

In their short communication, Walter et al. present a UAV-based semi-automatic system for the multi-temporal topographic monitoring of sediment dynamics in the source areas of a debris-flow basin. The study presents preliminary results on the erosion/deposition processes in the analyzed time period and explores the potential of the technique for investigating debris-flow hazards. The format of the short communication is suitable for this manuscript, the text is well written, and the presented system is definitely unique.

In addition to the comments of the other reviewers, on which I agree, I have few minor remarks that might be considered prior to publication. Section 5 Discussion and Conclusion sounds very optimistic for what concerns the future application of the technique in other contexts. I would suggest to briefly discuss also the potential limitations of the proposed system. For instance, I wonder to what extend the proposed technique is replicable in other locations given the installation, maintenance and data-processing costs. Regarding the assessment of debris-flow hazards in the aftermath of an event, the continuous monitoring of the unstable slope with ground-based systems (with radar, seismic but also photogrammetric sensors) is not, in general, more suitable?

Following this comment by Velio Coviello as well as the technical questions raised in the other referee comment and in the community comment, we will include more discussion on the system limitations. Some are related to power access, GPS, telemetry and other infrastructure, while others result from topography and geographical extent of the region of interest. While we agree that ground-based methods may be more appropriate in some cases, extended valleys quickly demand for satellite-based or aerial remote sensing. In this case our system offers some advantages, which we will further discuss in the revised manuscript version.

---

## Author Response (AR1)

Referee's comments are in dark blue.

**General comments**

This "Brief communication" presents a novel technique that enables the monitoring of sediment dynamics in remote terrains. It combines photogrammetric processing with UAV and was tested in the Illgraben debris-flow torrent, located in Switzerland. The topic is perfectly fitting with the ones proposed by NHESS and the contents are relevant for researchers and practitioners. I recommend the publication of the ms after some minor revisions.

- I suggest adding some information on recent studies applying UAV (or TLS) in torrential or fluvial areas (not related to Illgraben).

  In the discussion we now include two additional paragraphs about such studies, which frame the context of our contribution.

- The (preliminary) results, described between L125 and 131, should be extended and placed in a separate section. The results are brilliant and deserve a longer description. Not only related to the sediment dynamics, but also on basic (more technical) information like the pixel size of the DEM, which is missing.

  We included additional discussion on tributary activity but prefer to refrain from too much speculation, since we have not yet calculated sediment balances. However, this is planned in future studies, which we also now explain. To keep the paper structure simple, we did not split up the discussion into an additional section.

  Pixel size is given (10 cm).

If the above two points are not possible due to space problems, try to reduce other parts of introduction or discussion-conclusions.

- The description of locations like Illgraben mouth, channel outlet, upper catchment, head of the Illgraben channel, catchment outlet are not always clear. The authors may simplify them and add the most important ones in Figure 1A.

  Agreed. We further annotated Figure 1 and reduce closed terminology. In Figure 1 we labeled the upper catchment, the Illgraben mouth and outlet (we only use "channel outlet" now). We keep the word "head" as this seems standard for referring to a torrent's highest point.

**Specific comments:**

L30: "DURING debris flows" is not clear. The surveys were before and after debris-flow events, weren't they?

We reworded.

L73: width not with

Done.

L84-85: not totally clear, which was finally used in Illgraben (LAN or/and GSM).

We clarified.

L89: "1-inch Complementary Metal Oxide Semiconductor sensor" only expert may understand it. Please clarify.

This is a technical specification, which we deemed of interest to photogrammetry experts. The acronym "CMOS" is more widely used, but since we do not use this term again, we assume that we have to spell it out.

L125: Figure 2 (not 1).

Done.

Figure 2: Improve the design or layout (e.g. rotate the zoom boxes and make them larger). Clearly indicate the locations of the check dams.

Zoom boxes were enlarged, but we feel that they should be oriented the same way as the overview. Check dams are now indicated in the zoom boxes.

L140: You only mention flight time, but it would also be interesting having some information on the time consumption of the photogrammetric processing.

We now include the approximate processing time for our machine.

Community comments are in dark blue.

In this 'brief communication', Walter et al. introduce a new technique for (semi-) autonomous high-resolution monitoring of sediment dynamics in remote alpine terrain based on UAV remote sensing and photogrammetry. The presented system consists of a hexacopter and a base station that facilitates automatic recharging and acts as an operational relay between the UAV and a remote operator, who is in charge of the flight monitoring. From a technical and operational point of view, a remote operator would not be necessary within this framework, but supervision by a human is still mandatory for (autonomous) UAV missions beyond visual line of sight. The feasibility of the approach has been successfully demonstrated at the Illgraben debris-flow catchment in Switzerland and paves the way for (semi-)autonomous UAV-based monitoring in other contexts and terrains.

In addition to the comments of Reviewer 1, I have some (rather technical) remarks and questions that might be of interest for some readers:

1) A limited durability and replacement of some system components is mentioned in the text (L93-94). Which components were less reliable or failing and how often had the base station to be serviced during the summer months? The maintenance aspect would be especially important for autonomous operations in even more remote locations where the frequent replacement of components is difficult.

We followed this suggestion and implemented the changes we had proposed in the initial response to the reviews.

2) Had the remote operator/observer in the control centre in St. Gallen ever to intervene during the three-year period? Has the remote operator/observer full control over the UAV and what would happen in the case of an (unlikely) emergency (e.g. bird attack or curious people/animals approaching the base station during the survey or landing process)? I assume the base station is fenced. Is this correct?

We followed this suggestion and implemented the changes we had proposed in the initial response to the reviews.

3) Can you say anything about the precise and autonomous landing of the copter: is the RTK GNSS in combination with a vision-based tracking system used for this purpose?

We followed this suggestion and implemented the changes we had proposed in the initial response to the reviews.

4) Does the base station rely on an external power supply or is it connected to solar panels and a power station (not shown in Fig. 1)?

We followed this suggestion and implemented the changes we had proposed in the initial response to the reviews.

5) I'm aware that the selected manuscript type has a strict page limit, but I am missing a brief discussion on similar approaches/applications (i.e. semi-autonomous UAV monitoring) in the geosciences.

In the discussion we added two paragraphs discussing applications of UAV and terrestrial laser scanning and putting our study in the respective context.

Wording: Unoccupied Aerial Vehicle is increasingly used in the (geo)scientific community as a more neutral term for UAV (Joyce et al., 2021: https://doi.org/10.3390/drones5010021) and I therefore suggest to adopt it.

We changed the text accordingly.

Referee's comments are in dark blue.

In their short communication, Walter et al. present a UAV-based semi-automatic system for the multi-temporal topographic monitoring of sediment dynamics in the source areas of a debris-flow basin. The study presents preliminary results on the erosion/deposition processes in the analyzed time period and explores the potential of the technique for investigating debris-flow hazards. The format of the short communication is suitable for this manuscript, the text is well written, and the presented system is definitely unique.

In addition to the comments of the other reviewers, on which I agree, I have few minor remarks that might be considered prior to publication. Section 5 Discussion and Conclusion sounds very optimistic for what concerns the future application of the technique in other contexts. I would suggest to briefly discuss also the potential limitations of the proposed system. For instance, I wonder to what extend the proposed technique is replicable in other locations given the installation, maintenance and data-processing costs. Regarding the assessment of debris-flow hazards in the aftermath of an event, the continuous monitoring of the unstable slope with ground-based systems (with radar, seismic but also photogrammetric sensors) is not, in general, more suitable?

Following this comment by Velio Coviello as well as the technical questions raised in the other referee comment and in the community comment, we included a description of technical challenges that we faced during the autonomous drone deployment. We also now discuss limitations as well as advantages compared to terrestrial laser scanning.